# Multimodal Genealogy: The Capitol Hill Riot and Conspiracy Iconography

**Vittorio Iervese**

Department of Studies on Language and Culture, University of Modena and Reggio Emilia, 41121 Modena, Italy; vittorio.iervese@unimore.it

**Abstract:** The Capitol Hill riots on 6 January 2021 were an event of great importance not only because of their political and legal impact, but also because they allowed everyone to observe the symbols, images, masks, and other signs that were displayed in front of the cameras of many journalists and eyewitnesses. The iconography displayed on that occasion should not be dealt with as an extemporary invention but considered the result of a process of semantic and narrative accumulation produced in online and offline interactions. This article seeks to outline a theoretical–methodological framework of contemporary conspiracy images as multimodal forms of communication. Starting with images collected on Capitol Hill along with a corpus of online conversations that occurred on platforms such as Gab, in particular, between 2016 and 2021, examples of the dynamics of constitution of conspiracy images and their genealogy will be provided.

**Keywords:** conspiracy; iconography; multimodality; alt-right; hermetic semiosis

## 1. Introduction

On 5 January 2024, Donald Trump delivered the longest speech of his political career in Sioux (Iowa).[1] As is now his habit, the speech covered various topics ranging from local to international issues, alternating between personal anecdotes, mockery of opponents, exaltations of his cult of personality, and references to hidden plots. Several analysts considered this rally as a sort of compendium of Trumpian rhetoric and his strategy for building consensus (Joscelyn et al. 2024). Given that it did not fall on any ordinary date but on the eve of the third anniversary of the assault on Capitol Hill, Trump's speech could not but contain a reference to that dramatic and controversial event. In addition to expressing solidarity with the arrested individuals and confirming the accusation of alleged electoral fraud, Trump blamed infiltrators for the violent outcomes of that march (5 people dead) who allegedly deliberately triggered a chain of actions and reactions that led the situation out of control:

> "By the way, there was Antifa, there was FBI, there were a lot of other people there too leading the charge (...) You saw the same people that I did"

(Donald Trump at Sioux, Center Iowa on Friday, January 5)

This is not the first time that the thesis of the "conspiracy of the conspiracy" (Konda 2019) has been spread by some of the alt-right militants or by journalists and supporters of the former US President. Shortly after the events of 6 January 2021, when the dynamics of the events were still unclear, the first allusions to alleged leftist groups and undercover agents who unleashed violence within a peaceful and merely demonstrative protest began to circulate. The most interesting aspect of these hypotheses, repeatedly refuted by independent investigations (Gates 2024), lies not so much in what they assert but in how they seek to demonstrate their credibility. It is not surprising, in fact, that Trump's narrative attempts to overturn the evidence of facts, as the paranoid re-signification of reality's manifestations is a recurring and necessary strategy for maintaining the continuity of conspiracy theories

([Cassam 2019](#)). This is particularly necessary in the face of events that blatantly contradict the teleological framework on which conspiracy plans are based and could potentially refute their fulfillment. Specifically, the attack on the Capitol Building had been interpreted by many as the final manifestation of a plan with blurred outlines, which was supposed to lead to a rebellion by the "people" against a group of corrupt usurpers (the Deep State) and the reconquest of power. The march towards Capitol Hill and the entry into government buildings were supposed to represent the epiphany of a prophecy laden with expectations over the years. However, the reckoning did not occur as imagined, and instead, that event turned out to be a chaotic rally that ended dramatically with Trump's indictment for incitement. It is not surprising, therefore, that the response to this disappointment in conspiratorial expectations is a narrative that transforms failure into further evidence of a hostile and surreptitious plot that is even more complex and articulated than before. What is of interest here is the reference to documentary evidence made by Trump and his followers. In several instances during the 5 January 2024 speech, Trump invoked the probative effect of images, symbols, and other signs observed during the hours of the siege of Capitol Hill (e.g., "You saw the same people that I did"). Trump's statements were followed by numerous other interventions, including those of some prominent Republicans such as Mo Brooks ("Evidence growing that fascist Antifa orchestrated Capitol attack with clever mob control tactics"), Paul Gosar ("This has all the hallmarks of Antifa provocation"), and Matt Gaetz ("compelling evidence shows that Antifa had infiltrated the Trump-supporting rioters' ranks"). These statements also insist on "the symbolic context that gives meaning to social action" ([Gunnell 1968](#), p. 184). "Evidence", "hallmarks", etc. are terms that suggest a possible iconographic investigation that would incontrovertibly demonstrate the affiliation of certain individuals to certain groups who form part of a certain plot of domination and oppression. Trump's and his supporters' accusations have not been followed by detailed evidence, but it is interesting to note that in some cases, it is suggested that not only should evidence be collected but it is also worth studying the way in which certain symbols and images establish themselves and activate engagement processes.

This article aims to take up this provocation, to literally take the invitation that comes precisely from the exponents of the alt-right, not so much to debunk or dismantle certain conspiracy hypotheses but to attempt to navigate the field of political participation nourished by images, performances, and narratives. On the one hand, the statements of Trump and his followers make evident the centripetal and centrifugal forces of symbols ([Leib et al. 2000](#)), that is, the need to use symbols and images to aggregate an ingroup and create separation from the outgroup. On the other hand, the march and siege of Capitol Hill represent an event of enormous interest for its visibility because it allowed the observation of a theory of symbols, images, performances, masks, and other signs that emerged in front of the lenses of numerous journalists and eyewitnesses. The numerous recordings and photographs taken during those dramatic hours therefore allow for exceptional verification and analysis of the symbology and iconography of American political groups and movements based on conspiracy theories.[2] However, these symbols and images should be considered the result of a process of semantic and narrative accumulation that occurred in online and offline interactions. Symbols and images should therefore be considered not only for their intrinsic meaning but also as "visual cues" ([Trujillo and Holler 2021](#)) that position participants and contribute to communication.

## 2. Towards a Multimodal Genealogy

In the past, some scholars of undisputed relevance have referred to a possible "genealogy of the image" (e.g., [Warburg 1998](#); [Didi-Huberman 1990](#)). These references point to approaches and objectives that are significantly different from each other to the extent that it is not possible to separate a coherent theoretical and methodological framework. However, it is possible to identify a common trait among these references, not so much because they attempt to reconstruct a relationship of descent between the copy and original but because they question how images become part of and operate in certain "environments"

or "systems of meaning" (Manghani 2013). These environments or systems of meaning resemble those "finite provinces of meaning", which, according to Schütz (1945), allow "common sense" to become a device for organizing social interaction. The concept of "finite provinces of meaning" focuses on the ways in which meaning is structured and articulated, that is, how cognitive maps are configured and allow individuals to organize experiences according to specific cognitive styles and to recognize the situation in which they find themselves inserted. Each province encompasses a set of experiences to which a similar nuance of meaning and a comparable tenor of sense are attributed. From this perspective, images would be subject to and object of processes of accumulation of meanings that, over time, structure and organize (Iervese 2024). The provinces of meaning are described as "finite" in the sense that they are organized, not because they are closed or given once and for all. On the contrary, images evoke and help define finite provinces of meaning with mobile and porous boundaries. It is therefore from the identification of the spaces and practices of signification that one can attempt to describe a genealogy of images. As W.J. Mitchell states regarding the term image:

> If we begin by looking, not for some universal definition of the term, but at those places where images have differentiated themselves from one another on the basis of boundaries between institutional discourses, we come up with a family tree... [which] designates a type of imagery that is central to the discourse of some intellectual discipline. (Mitchell 2013, pp. 4–5)

However, this genealogical tree is articulated, rhizomatic, with multiple filiations, grafts, and hybridizations—"thought-images" (Denkbilder) that produce "constellations of meaning" (Benjamin 2010) capable of breaking the usual coordinates of time and space and therefore also classical genealogies (Weigel 2015). Therefore, each image:

> [is] not only connected forward and backward in an "unilinear" development [but] it could only be understood by what it derived from and by what it contradicted. (Gombrich 1986, p. 284)

In this way, a dialectical method is outlined in which objects are not classified according to the historical narrative of art but rather consider "the contrasts, similarities, tensions, and anachronisms between them" (Blassnigg 2009, p. 42). This approach has significant theoretical and methodological consequences. The main one is that, in the words of Didi-Huberman (1990), it questions the "rhetoric of certainty" affirmed over the years by disciplines that deal with the contents of images. It is not a matter here of opposing one discipline to another but of affirming that: "the image is important not so much for what it shows, but for what it relates and for the form of that relationship itself" (Dinoi 2012, p. 12). Images are the result of this network of relationships, for some they are even actors with their specific agency or actancy (Latour 2005). Consequently, images should not be studied in isolation, as circumscribed and isolated objects, but as sets of practices that not only vary in use but also in their meaning, depending on the contexts and communicative processes in which they are inserted. Here, it is not only a matter of recognizing that "We never look at just one thing; we are always looking at the relation between things and ourselves" (John Berger 1973, p. 7) but also that images activate a cognitive and social experience that is not only visual but multimodal. According to social semiotics, a modality is defined as "socially shaped and culturally given semiotic resource for making meaning" (Kress 2010, p. 79). With multimodality, therefore, reference is made to the coexistence, within communicative processes, of visual, auditory, linguistic, gestural, performative, material, etc., modes that favor the positioning of participants, the interaction between them, and the shared elaboration of meanings.

According to traditional scholasticism and following classic theories of representation, we have been accustomed to seeking the content of an image, asking questions such as "What does it mean?", "What does it contain?", "What is inside?", and "What does the author want to express?" As early as the end of the last century, many studies emerged (Curtis 2010; Mitchell 2013) that generated an epistemological inversion in the study of

images, replacing previous questions with others such as "What are you asking? What are you looking for?" and "Where does that image want to lead me?" Images are thus understood as a complex interaction between visuality, social systems, institutions, discourses, bodies, and figurativeness. Moreover, images become part of an inextricable network of references between immaterial images (images) and material images (pictures) that are produced, exchanged, and acted upon daily (Iervese 2016). We can refer to this network as an imaginary in which the iconic, narrative, and performative dimensions blend and feed off each other. Every image depends on an imaginary that, in turn, helps to construct it. And it is thanks to this complicity that images become alive and capable of constructing a discourse that has real effects.

The numerous symbols and images that appeared during the Capitol Hill Riot (CHR) on 6 January 2021 can be considered as forms of multimodal communication that exert an aggregative and political force through at least three dimensions that connect images and symbols to the (1) narrative dimension; (2) performative dimension; and (3) discursive dimension. In the first case, reference is made both to the evocation of stories and characters from the past and to possible future narratives. More precisely, the prophetic and teleological forms on which every conspiracy theory relies converge the stories of the past and those of the present to prove the existence of a future destiny. In this sense, the connection between the iconic dimension and the narrative one favors the self-evidence and the incontrovertibility of facts and events, as well as the confirmation of cultural positions and presuppositions (Baraldi et al. 2021).

The second connection is the one that allows images to play a role in interactions and communications. The images at Capitol Hill were not only seen but also acted upon, worn, incorporated, exhibited, idolized, etc. On the one hand, images are accompanied by different modes of expression (written, oral, behavioral, etc.), and on the other hand, images themselves become social actors; they assume their own autonomy within social interaction. Not only are things done with images, but images suggest and inspire one to do things.

The discursive dimension introduced in the third connection is to be understood in Foucauldian terms, that is, as the set of procedures that define a regime of truth. The images used by conspiracy groups establish a connection with a discourse related to power and control institutions that subverts and, in some ways, instrumentally uses the method introduced by Foucault. One of the most recurring attempts of conspiracy strategies is precisely to attack and contest the systems of delimitation and control of discourses in the name of a supposed freedom of thought and speech, as well as unveil a truth hidden by established powers. Just think of the so-called exclusion procedures within which Foucault (1971) inserts the mechanism of the interdict, the partition (or partage) between reason and madness, and the opposition between the true and the false. These mechanisms are, more or less implicitly, instrumentally used and even overturned by those who adhere to conspiracy theories. Paraphrasing Foucault's words, the groups dedicated to unveiling and fighting alleged conspiracies affirm that (a) anyone can talk about anything; (b) the discourse of the madman cannot circulate like that of others; and (c) the will to unveil a hidden truth consists of opposing the opposition between true and false as imposed by those who conduct plots or maneuver in the shadows.

*Methodology*

The images and symbols accompanying the events at Capitol Hill and the attack on the Capitol Building delineate a multimodal social field, thus being complex and ambiguous. The following paragraph will analyze some of these images and their relationship with the narratives, performances, and discourses in which they are embedded and which contribute to their construction. Before delving into the analysis, it is useful to provide some insights into the methodology employed. This article draws inspiration from a broader and more comprehensive study on the multimodal communicative processes defining the conspiracy visual field. Consistent with the aforementioned, the analysis

does not solely focus on the "content" of the image but rather on how images are part of interactions and communications.

Initially, iconic materials or visual cues were collected from the field, involving the observation and cataloging of symbols and images associated with conspiracy groups within the broader American right-wing spectrum. Specifically, movements proposing radical right-wing ideologies alternative to traditional American conservatism, such as the alt-right, were isolated. If all those who were present on Capitol Hill can be considered supporters of President Trump, it would be methodologically incorrect to consider all the images and symbols displayed there as part of conspiracy thinking (Hawley 2017). Therefore, after a careful identification of the main symbols and images recognizable at Capitol Hill, only those used by groups identifying with conspiracy theories or self-defined as conspiracists in their communications on websites, forums, and social networks were further investigated.

To this end, various online discussion platforms such as Gab, 4chan, 8chan, Reddit, and Parler were reviewed between 2016 and 2021. The choice to focus on this time frame is motivated by two specific reasons: the 2016 US presidential elections, which witnessed Donald Trump's ascent, and the subsequent intensification of activism among alt-right groups and related conspiracy narratives. Trump's presidency further galvanized these groups, leading to their consolidation, expansion, and organization across numerous discussion forums, among other platforms. The case of Gab serves as an exemplary illustration in this regard. Conceived by Andrew Torba, then a twenty-five-year-old who was the CEO of "Automate Ads", a start-up based in Scranton founded in 2011 to promote advertising campaigns on Facebook and Google, Gab presents itself as a microblogging platform similar to Twitter. Torba, a staunch supporter of Donald Trump, expressed concerns over the censorship faced by some Republican accounts on social media platforms like Facebook. Consequently, Torba decided to create a new platform, free from any form of censorship, with the aim of promoting "free speech" and alt-right ideology (ADL 2022). Gab was launched in beta on 15 August 2016, before its full release in May 2017. The trial phase of Gab occurred during the electoral period, functioning as a kind of neutral zone for alt-right activists and sympathizers seeking interaction and aggregation without constraints or controls.

Over the years, Gab has become a focal point for Trump supporters, conspiracy theorists, and American neo-Nazis. Public attention on Gab peaked at two crucial moments in recent US history. The first was the Pittsburgh synagogue shooting on 27 October 2018, perpetrated by Robert Gregory Bowers, who killed 11 people and injured 6. Bowers had been highly active on Gab with violent and antisemitic posts made prior to the attack (Shannon 2018). Just hours before the shooting, Bowers declared his intention to turn his words into actions through this post on Gab[3] (Figure 1):

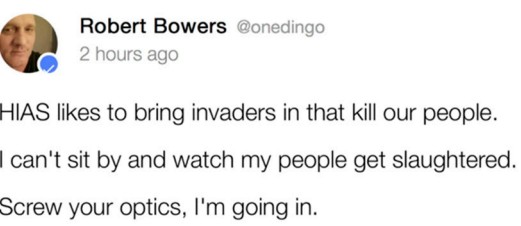

**Figure 1.** Screenshot taken from the Gab platform (27 October 2018).

Gab regained public attention in 2021, both because the first images and videos of the Capitol Hill attack circulated on that platform and because, according to some investigations, protesters used Gab to organize themselves before and during the siege (Frenkel 2021). Thanks to specific APIs provided by the platform itself (Gab Scraper) or non-profit research projects like Distributed Denial of Secrets (DDoSecrets), it was

possible to download data and metadata from their respective social media and analyze the conversations and content posted.

The research was based on recordings made by various broadcasters and journalists on the spot, but also on videos posted on social media by users participating in the riot. Thanks to the collection carried out by ProPublica, a project of "Investigative Journalism in the Public Interest",[4] it was possible to view more than 500 self-produced videos uploaded in the hours of the attack directly on the Parler platform. We chose to pay attention to these materials in addition to those circulated in the media, both because they are the fruit of the self-representation of the rioters themselves and because they were not subject to editing or intermediation. The symbols detected in these videos were compared with the Hate Symbols Database constructed by the Anti-Defamation League, which contains more than 250 different symbols considered to be in some way related to racist, supremacist, conspiracy speech. The expanded corpus therefore includes "a toxic mix of groups from across the far-right spectrum". (Miller-Idriss 2022). As the ADL itself also states:

> All the symbols depicted here must be evaluated in the context in which they appear. Few symbols represent just one idea or are used exclusively by one group. For example, the Confederate Flag is a symbol that is frequently used by white supremacists, but which also has been used by people and groups that are not racist'. (ADL 2019, p. 60)

For this article, about 50 symbols appeared in different forms and on different mediums (flags, stickers, t-shirts, gestures, tattoos, etc.). Among these, especially those that expressed their aggregative and communicative power were analyzed, i.e., those on which conversations recorded on online discussion platforms such as Gab. Specifically, 85 conversations of varying length on 15 different symbols were analyzed. This was retrospective, somewhat genealogical research, insofar as the communications analyzed allowed us to collect data on the accumulation of meanings attached to the images and symbols used as vehicles for conspiracy theories. Once again, it is important to specify that the relevant issue is not so much about finding information on the content of the image but understanding how these images become part of communicative and signification processes that lead to collective identification. Due to space constraints, it is not possible to report the conversations and communicative processes analyzed in this article, which will only be mentioned in relation to the cases presented in the next paragraph.

**3. The Conspiracy of Images**

In Section 2, some peculiarities of the images and symbols used in the vast realm of American conspiracy theories were briefly mentioned. Some of those characteristics could be applied to many symbols and images of other political movements, as well as to the forms of constitution of political imagery in general. Symbols are not mere accessories to political inquiry but are "the phenomena themselves of social [political] inquiry" (Gunnell 1968, p. 185). Symbols are an integral part of how we perceive and understand political life and its activities. In fact, according to John Gunnell, "the social scientist must aim to illuminate the symbolic context that gives meaning to social action" (ib. 184). It is therefore pertinent to ask whether specificities of the use of images by contemporary conspiracy groups can be identified and whether there is a difference compared to other political iconographies.

*Ambiguity, Hybridization, Adaptability*

In general, most of the images and symbols upon which conspiracy narratives and communications are based usually possess characteristics that predispose them to mutable and "customizable" use. In this sense, they differ significantly from other symbols and images that characterized political identities of the last century. If one carefully observes the heterogeneous iconographic theory displayed at Capitol Hill, one notices numerous symbols and images that can be traced back to classic and historically identifiable political groups. Even the many flags of nations or Confederate states should not be interpreted

as evidence of the participation of some territorial delegation but rather as allusions to narratives and events to which those flags refer. It is interesting, in this regard, to note that among the flags waved by the crowd gathered at Capitol Hill, there were many of foreign nations, each of which could be linked to identity conflicts or contested narratives. Take, for example, the flag of Georgia (understood as the Caucasian nation and not as the federated state of the USA, Figure 2).

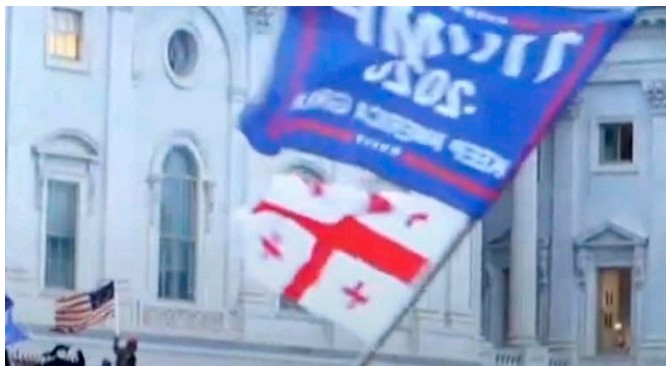

**Figure 2.** The flag of Georgia. Still image of the footage taken during the CHR.

That flag, commonly known as the five-cross flag, has been misinterpreted by many inattentive commentators as a nationalist claim or even as a glaring error to be stigmatized and ridiculed. However, from various online conversations recorded on Gab (Gab 1, 2018, 2019, 2021),[5] a clear reference to the Jerusalem cross and therefore to the Militia Christi, or the religious and military–chivalric order founded to defend Christian values from heresies, emerges. In other posts, the flag of Georgia is associated with the motto "Deus lo vult" used by Peter the Hermit around 1096 to organize the so-called People's Crusade, a spontaneous and unorganized precursor to the First Crusade (Flori 1999). These posts and the ensuing discussions focus almost exclusively on self-organization, on the people rising up and needing to "take action" (Gab 2, 2016–2021) to "defend ourselves from those who want to annihilate us" (Gab 3, 2021). This allows for linking a narrative of the past to a conspiracy theory such as the Great Replacement or genocide by replacement (Camus 2012).

It is important to specify that the instrumental use of historical narratives or figures does not presuppose a deep and correct knowledge of the historical events evoked; on the contrary, errors of interpretation, interpretative distortions, and forced interpretations are frequent if not systematic. The communications activated from the flag of Georgia confirm instead the use of "finite provinces of meaning" adaptable and usable to frame, make sense of, and take a position on contemporary phenomena. The perimeter of this area of meaning is delineated by some general concepts such as "defense" (of Christian roots), "supremacy" (of the white West), "heroism" (of militancy), etc. The space of latency between these "pathosformeln" (Warburg 1998) is crossed by narratives, performances, and discourses made possible precisely through the mediation of image and symbol. Therefore, it is not sufficient to understand the "content" of the image or establish the truthfulness of the association between image and reality. More importantly, on the one hand, it is necessary to reconstruct the processes of semantic accumulation that allow that image to guide social interactions. On the other hand, it is equally important to observe how these images facilitate and predispose themselves for further communications and hybridizations.

On the topic of the flags featured during the CHR, many are presented in their modified version through additions, omissions, their small shapes, and color transformations designed to emphasize reference to a conflictual history or controversial memory. The tendency to personalize and appropriate certain symbols by modifying them is one of the constants in conspiracy movements. In this way, some historical narratives are reconfigured to their own benefit, or collective identities are carved out in their likeness. Examples abound; in addition to the many versions of the Confederate Flag that exploit the

original reference to the Civil War to become an expression of discourse focused on White Supremacy (Leib et al. 2000), the case of the "Blue Lives Matter" movement in support of US police forces and in open antagonism with the claims of the African American population is interesting. The Thin Blue Line Flag (Figure 3), used by the Blue Lives Matter movement, replaces a white stripe of the US flag with a blue one in reference to the uniforms of police forces. This small variation is sufficient to transform narratives and possible identifications. The centripetal force of the symbol is transformed in this case into a line of conflict demarcation towards other claims and affiliations (Black Lives Matter). As proof that these variations are very common, it is enough to remember that there are also the Thin Red Line Flag, the Thin Green Line Flag, and the Thin White Line Flag connected respectively with the jobs of firefighters, military personnel, and paramedics.

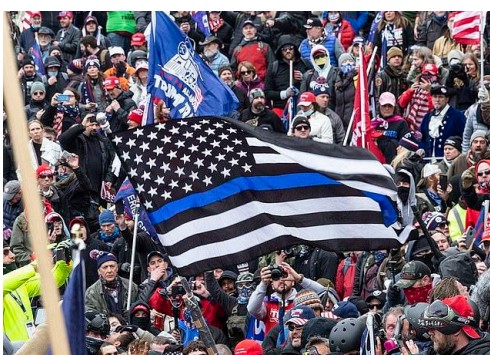

**Figure 3.** The Thin Blue Flag. Still image of the footage taken during the CHR.

In general, most images and symbols observed during the CHR are 1. ambiguous; 2. hybrid; 3. adaptable. Ambiguity refers to the relationship between denotation and connotation, a central distinction in semiotic studies (Leone et al. 2020). The use of images and symbols by the protagonists of the CHR does not indicate but suggest, does not say but allude, does not clarify but imply. This is a contradictory quality because on the one hand, it seeks to maintain the secretive nature typical of clandestine groups that only address their members or initiates. On the other hand, these symbols and images are designed to be spreadable (Jenkins et al. 2013) and viral (Berger and Milkman 2012); they become part of a merchandising plan with evident commercial interests, follow social branding strategies, etc. However, ambiguity is necessary to seamlessly navigate the space between the scene and backstage or, otherwise, the public and non-public space. Ambiguity also serves to give a mimetic and sometimes parodic character to symbols and images that are prosecutable and condemnable. In other words, ambiguity and the metamorphic use of symbols also serve to protect against potential accusations or charges from those who associate the display of certain symbols with racist or hateful messages towards other groups. The Anti-Defamation League's Center on Extremism states its mission on its homepage: "We track extremist trends, ideologies, and groups across the ideological spectrum. Our staff of investigators, analysts, researchers, and technical experts strategically monitor, expose, and disrupt extremist threats". Moreover, among its main activities is also that of "providing an overview of many of the symbols most frequently used by a variety of white supremacist groups and movements, as well as some other types of hate groups", with the awareness that these symbols are ambiguous because they spread while concealing themselves.

A case of easy understanding is the transformative appropriation made by militants and sympathizers of white supremacist groups of the gesture conventionally used to communicate that everything is "ok" (the index finger and thumb united to form a circle; the remaining fingers extended upwards). Militants have spread the habit of making this gesture by showing the back of the hand instead of the palm. In this way, the gesture alludes to the letters W and P of White Power but allows it to not openly contradict its original meaning, which is instead the acronym for Oll Korrect, a corruption of "all correct" in vogue in East Coast cities in the first half of the 19th century. This all started during

the campaign for the US presidential elections in 2016 when the gesture began to spread among some well-known far-right pundits like Milo Yiannopoulos, but it was consecrated in a campaign that emerged in 2017 on channels like 4chan and Gab. In this campaign, unequivocally called "Operation O-KKK", all the evidence is present of a conscious use of ambiguity and ambivalence as a method to bring about the semantic inversion of the common use of a gesture unwittingly used by a multitude of American citizens.

> The "okay" gesture hoax was merely the latest in a series of similar 4chan hoaxes using various innocuous symbols; in each case, the hoaxers hoped that the media and liberals would overreact by condemning a common image as white supremacist. In the case of the "okay" gesture, the hoax was so successful the symbol became a popular trolling tactic on the part of right-leaning individuals, who would often post photos to social media of themselves posing while making the "okay" gesture. (https://www.adl.org/resources/hate-symbol/okay-hand-gesture, accessed on 9 May 2024).

It is evident in this example how the symbol is performed and used as an ambiguous, hybrid, and adaptable communication medium. This practice is fully consistent with the systematic exercise by conspiracy groups of what we might call a rhetorical strategy of double entendre "in which every symbol or image can be read as a sign that carries, behind its conventional meaning, an arcane meaning that betrays the existence of an invisible plan" (Pannofino and Pellegrino 2021) and which can only be understood by those who know how to read and use the appropriate symbols.

This example also brings to light a central aspect of conspiracy strategies; rather than asserting content, the goal is to provoke a chain of actions and reactions. Terms like "owning the libs" (Perticone 2018), "triggering the libs", and "melting snowflakes" (Coppins 2018) indicate strategies used by right-wing activists in America to dominate political communication by putting liberals in a position where they often have to respond emotionally and in a discomposed manner to the provocations circulated for this purpose. The case of the "ok" hand sign is emblematic; some users of 4chan and Gab spread the false idea that it was a new symbol of white supremacism, more to put pressure on or to create confusion among those who did not share that vision than for convinced supremacists, who started using it sincerely, knowing they could always defend it as a well-known and harmless gesture. These trolling strategies should therefore not be considered for what they affirm in individual communication or speech but, above all, for what they activate in communication flows.

The ambiguous nature of conspiracy symbols and images is also due to their ability to generate hybrid images, often the result of a bricolage of forms, cultural inspirations, and different repertoires. In conspiracy images, historical references are mixed with fantasy inventions, artistic canons with media products, vernacular narratives with cinematic references, etc. In several rallies, Donald Trump attacked Joe Biden with various arguments concluding by stating that he is "a compromised president who is leading our country to hell. He is a Manchurian candidate!" (Parry-Giles and Barney 2020). In this way, a speech with debatable but politically oriented arguments ends with a metaphor taken from the book from which two films have been made, titled *The Manchurian Candidate*. The metaphor of a candidate controlled by external forces to attack the heart of the United States has been repeated several times by Trump and his supporters, leading to a series of multimodal items (e.g., memes) spread across various social platforms. Despite different historical references, over the years, the idea of a President who has been "brainwashed" to comply with the wishes of an elite plotting in the darkness has become a paradigm that accompanies the multiple paranoid narratives of American conspiracy groups (Kim 2010).

Another example can be provided by the Proud Boys, an extreme right-wing group founded in Canada in 2016 whose members define themselves as "Western chauvinist[s] who refuse to apologize for creating the modern world" and profess, among other things, a radical libertarianism ("give everyone a gun"; "end welfare") and traditional gender roles ("venerate the housewife") (Kenes 2021). This "misogynistic, Islamophobic, transphobic

and anti-immigration" (McBain 2020) group owes its name to a song titled "Proud of Your Boy", contained in the first version of the Disney movie *Aladdin* (Kenes 2021). In this case, the hybridization is even more surprising—on the one hand, because it draws from an animated work aimed at children and a family audience to create the identity of a supremacist and violent movement. On the other hand, the incongruence between the values of this group and the character of Aladdin, a young Middle Eastern man who lives by theft and deception, may be astonishing. But hybridization should be understood as a practice of parody, remix, and mimicry that combines materials and sources of different origins, appropriating certain parts to reassemble them in a way that is useful for a thesis. Thus, Biden can easily be described as a Manchurian Candidate and Aladdin has become the inspiration for an organization composed of white, racist males.

These multimodal items are designed to be adaptable and modifiable over time; that is, they are conceived as a modular structure depending on one's interests and contingencies. In these communicative strategies, images are fundamental more for what they relate to or enable one to (re)produce than for what they show. Hypericons (or metapictures) is the term that best indicates the redundancy and reproductive capacity of certain contemporary images (Mitchell 1986). Hypericons are political and aesthetic assemblages, images that generate other images that create effects of reality. Hypericons aim to construct reality rather than represent it. An icon is two things in one; it is simultaneously an image and an idea, a sign and a symbol. The icon indicates the life space of an image that goes beyond that of its referent. A hypericon does something more; it provides instructions for the gaze, suggests ways of seeing, and introduces practices for using the image.

Hypericons provide a figurative nucleus, capable of assuming a generative function of representation. The value of hypericons lies not so much in the fact that they are beautiful, original, or unique in their kind. On the contrary, hypericons assume value and power precisely because of their ability to transcend their material support (the picture) and the referent of the image (the subjects, the event), thus being able to move from one medium to another, from one context to another. In this sense, the principal model of image use by contemporary conspiracies is the meme, understood as an ambiguous, hybrid, and adaptable multimodal item (Harbo 2022). Continuing to consider memes as a product of youth subcultures or as just another niche becoming mainstream does not help to understand the novelty they introduce into contemporary multimodal communication. Memes are communicative tools designed for reproducibility and virality thanks to their affordability. The concept of "digital affordance" can be summarized as "what an individual or an organization with a particular purpose can do with a technology" (Majchrzak and Markus 2013) where what "can be done" also implies what can be added and modified. Conspiracy iconography is therefore ambiguous, hybrid, and adaptable to promote engagement by users who become prosumers. If twentieth-century political symbols and images inspired respect and reverence, and were therefore considered untouchable, contemporary symbols and particularly those of conspiracy are by their very nature mutable, thus willing to be used even in an ironic or playful manner. In this way, these images spread even among those who do not share certain political positions or conspiracy theories to which they allude.

## 4. "Join or Die"

The redundancy of symbols and images present in the footage of the Capitol Hill Riot (CHR) offers numerous points of analysis and avenues for deeper investigation that cannot be fully explored in this essay. However, it is important, based on the considerations made in the preceding paragraphs, to succinctly provide examples of how the multimodal communication of conspiracy groups utilizes images as aggregators of both members and arguments. The degeneration of hermetic semiosis, inherent in every conspiracy, relies on the ability to convincingly assemble what is separate and disconnected. Singular events and facts are "recomposed, brought together in the light of a 'plan', so that a (phantasmatic) coherence can be restored to them" (Solinas 2023, p. 13).

Among the peculiarities of the images mentioned in the preceding paragraph is their ephemeral nature and instrumental purpose for a contingent aim (e.g., fueling a controversy, attacking a personality, etc.). However, it is not always clear how these images contribute to aggregating what is separated. While it is essential to understand the communicative use of the countless images circulating online, it is crucial to inquire about those that become icons and symbols of political aggregation, i.e., those with both centrifugal and centripetal functions, as mentioned earlier (Leib et al. 2000).

A first set of images recorded at Capitol Hill that have become prominent on digital platforms such as Gab are entirely based on fictional narratives, arising from playful interactions or trolling strategies. These images are of great interest for understanding contemporary mythopoesis dynamics. A well-known and striking example is that of the Kekistan flag, a fictitious country where political correctness is decried.

If the flag graphically resembles the Nazi battle flag (Figures 4 and 5), the rest of the narrative framework is devoid of concrete historical references and adherent to reality.

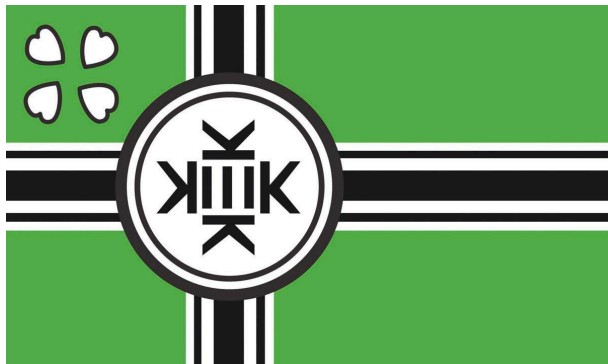

**Figure 4.** The Kek flag. (https://en.wikipedia.org/wiki/File:Flag_of_Kekistan.svg, accessed on 9 May 2024).

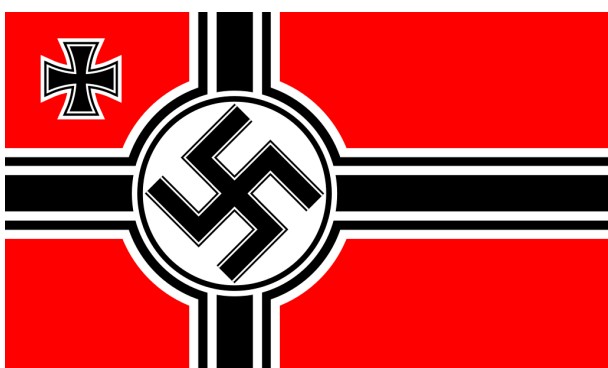

**Figure 5.** War Ensign of Germany (1938–1945). (https://en.wikipedia.org/wiki/File:War_Ensign_of_Germany_(1938%E2%80%931945).svg, accessed on 9 May 2024).

The term "kek" originates from the world of video games, and among users of 4chan and Gab, the false idea spread that it refers to the name of an ancient Egyptian deity with the body of a man and the head of a frog. Parodies and images begin to emerge, borrowing from an existing animated character like Pepe the Frog, who soon becomes the protagonist of countless memes with racist and conspiratorial undertones, or simply sarcastic ones. Pepe thus transforms into the champion of uncensored language and, therefore, of freedom of expression; "memetics" is described as a communicative strategy to overturn the established order (Gab 5, 2017, 2019). The success of this image, assuming a narrative, performative, and discursive dimension, is surprising, especially considering that there is no attention to historical coherence or the plausibility of what is disseminated; each

product or result is, in this case, subordinate to the process. This collection of multimodal communications left to the free initiative of online communities, often chaotic and without a precise program, is the basis of what is termed "new conspiracy without theory" (Muirhead and Rosenblum 2019) or "skeletal conspiracy" (Solinas 2023).

Another set of images presents greater backward- and forward-looking complexity. In these cases, it is possible to identify a genealogy from a historical event that serves as an origin but not necessarily as a consistency parameter. For example, a movement of growing importance like VDare declares its inspiration from the figure of Virginia Dare, the first English child born in 1587 in the New World. The disappearance of Dare and over 100 other settlers on Roanoke Island, located in present-day North Carolina, remains a mystery to this day. Interest in the Lost Colony has never waned over the centuries, and over time, it has become a persistent symbol for supporters of the replacement theory. In fact, among the various possible reconstructions of that event (Miller 2001; Parramore 2001), one has prevailed over the others, mixing historical sources with legends and fanciful suppositions. It is presumed that the colony was either exterminated or taken prisoner and then dominated by Native Americans (Figure 6).

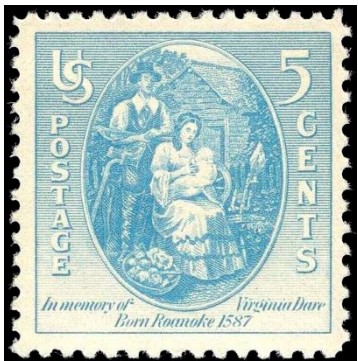

**Figure 6.** Postage stamp depicting Virginia Dare (child in middle). https://en.wikipedia.org/wiki/VDARE#/media/File:Virginia_Dare_5c_1937_issue.JPG, accessed on 9 May 2024.

From this hypothesis, it is easy to transform the Lost Colony into evidence of an impending "white genocide", a term referring to a racist conspiracy theory that essentially argues that white people are being systematically replaced by non-whites, for example, through an influx of immigrants. On the homepage of the VDare website, it can be read that "Virginia Dare became romantically entangled with an Indian sorcerer who transformed her into a white doe. It is from that image that we derive the VDARE.com logo" (https://vdare.com/, accessed on 9 May 2024) (Figure 7).

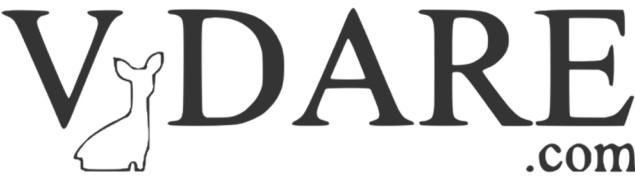

**Figure 7.** Vdare logo (https://en.wikipedia.org/wiki/VDARE#/media/File:VDARE_logo.svg, accessed on 9 May 2024).

As Walter Benjamin (1997) argued, history disperses through images and aggregates into narratives. In this case, the white fawn becomes the image that evokes a narrative, which, in turn, evokes a discourse. Also from the VD website, one can read this conclusion: "And it is in the name of Virginia Dare herself that we defend the traditional American community and give it voice. We cannot allow the Lost Colony to prove analogous to America itself" (https://vdare.com/ accessed on 9 May 2024).

Another similar but more famous example is that of the Gadsden Flag, which exists in various versions and represents a true case of transmedia diffusion, as well as multimodal expression. The Gadsden Flag typically features an image of a rattlesnake on a yellow background with the inscription "Don't Tread On Me" (Figure 8).

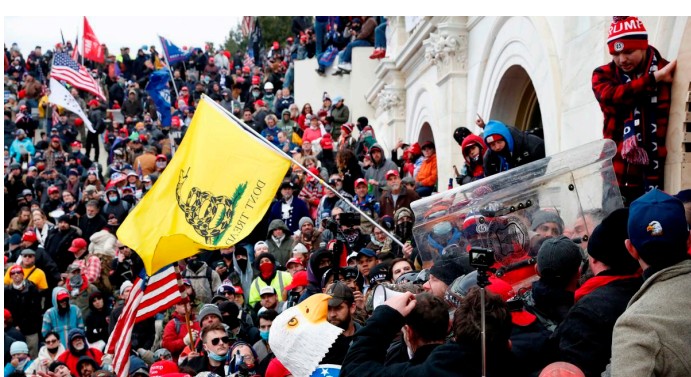

**Figure 8.** Gadsden Flag. Still image of the footage taken during the CHR.

The origin of this symbol can be traced back to an article by Benjamin Franklin dated 9 May 1751, titled "Felons and Rattlesnakes" (Labaree 1961). In this article, which reads like a political diatribe with satirical undertones, Franklin laments the continuous deportation of convicts and felons to the English colonies in America. In response, Franklin proposes sending rattlesnakes to England in exchange for the convicted prisoners sent to America ("Rattlesnakes seem the most suitable returns for the human serpents sent us by our Mother Country") (Labaree 1961, p. 132). The article was accompanied by an illustration depicting the colonies as a segmented snake with the caption "Join or Die", (Figure 9) along with the initials of the main states involved in this appeal.

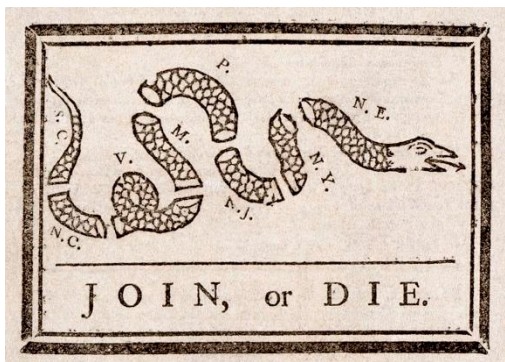

**Figure 9.** "Join or Die" (https://it.wikipedia.org/wiki/Join,_or_Die#/media/File:Benjamin_Franklin_-_Join_or_Die.jpg, accessed on 9 May 2024).

At that time, the superstition was widespread that if the segments of a snake were reassembled before sunset, it would come back to life. This was undoubtedly one of the inspirations for the illustration. The original perimeter of the area of significance, however, is traced by concepts such as "independence", "liberation" (from oppressive power), "unity" (strength in unity), "closeness to the land", and a series of characteristics that Franklin attributes to the identity of the inhabitants of the colonies, borrowing them from those of the rattlesnake:

1. Vigilance (always alert and watchful);
2. Magnanimity (does not attack out of hostility but only to defend itself);
3. Courage (if called upon to defend itself, it does not hesitate);
4. Unity (the rattle works only if the parts are connected to each other);
5. Circumspection combined with readiness to attack (teeth are hidden);

(Isaacson 2005, pp. 264–66).

Once the fighting began during the Revolutionary War, the newly formed Marines were seen with drums painted yellow with a coiled rattlesnake above the inscription "Don't Tread On Me". Franklin would have collaborated with Christopher Gadsden, representative of South Carolina and colonel of the Continental Army, to create the Gadsden Flag with the same design. Gadsden presented the flag to the Commander-in-Chief of the Navy, who accepted it as one of the "battle flags". The image of the rattlesnake was thus used as a symbol of the American colonies from the beginning to the end of the War of Independence. At the same time, the snake warning its enemies began its long journey appearing in cartoons, newspapers, and coins, as well as on other media such as flags, shirts, buttons, cemetery monuments, etc. (Ruppert 2015). The image became a hypericon and produced a series of other secondary images, hybridizing with other narratives, performances, and discourses. The Gadsden flag has made numerous appearances in popular culture, particularly in post-apocalyptic stories, becoming a symbol of libertarianism, described as conservative on economic issues (economic liberalism) and liberal on personal freedom (civil libertarianism), often associated with a foreign policy of non-intervention.

> In recent years, the Gadsden flag has become a favorite among Tea Party enthusiasts, Second Amendment zealots, really anyone who gets riled up by the idea of government overreach. It's also been appropriated to promote U.S. Soccer and streetwear brands. And this reflects a deeper question, one that's actually pretty compelling: How do we decide what the Gadsden flag, or indeed any symbol, really means? (Walker 2016)

This apparent impossibility to determine the true meaning of the Gadsden Flag is actually a common phenomenon in contemporary images that has become part of multimodal forms of communication. Thus, the perimeter of the original area of significance of the Gadsden Flag, from defending the freedom of a new form of citizenship in the American colonies, extends to include theories about immigration as a conspiracy against the USA. Similarly, reference to the magnanimous force of Franklin's interpretations makes way for the violent aggression with which a couple in Las Vegas in 2014 first killed three people and then laid out the Gadsen Flag and one with the swastika on their bodies (Berman 2014). In this way, a new teleological thought is imposed that does not concern itself with the high rate of inconsistency and contradiction but is oriented towards a practice that privileges assembly and hybridization, syncretism and invention, diffusion and convergence. Franklin's motto, "Join or Die", thus takes on a double meaning—one that invites one to aggregate, to participate, and to act as a mass but also one that suggests a practice of syncretism of images and narratives. Contemporary processes of political identification also pass through a continuous multimodal bricolage that does not confront the reality of facts but aspires to create and recreate them.

## 5. Conclusions

This paper, far from being exhaustive, has tried to provide an example of multimodal analysis and an attempt to enrich the field of research that lies in a complex and articulated area in which historical, sociological, philosophical, and political studies converge. The focus on images, understood not only as representations or iconic compendiums of the world but also as an intrinsic part of discourses, narratives, and social practices, is fundamental to the evolution of these interdisciplinary studies, especially in view of the importance acquired by the visual in contemporary performative communication. The contemporary image presents itself as both product and process, as something to look at and, at the same time, as something to act upon.

The recent debates on the fake, on the iconosphere, on the substitution of the real, on automation, etc., which are purged of an annoying as well as superficial apocalyptic or, on the contrary, salvific rhetoric, inform us of how images are no longer a sign fact but an ontological occurrence. It is precisely the investigation of this ontological status that

differentiates recent *visual culture* studies, of which this article can also be considered part, from the semiotic studies of the last century. The "new" digital images play a central role in the negotiation of multiple sets of realities and discourses of truth because they never stand alone but have to be considered in connection with software devices, computational techniques, networks, platforms, control systems, etc. Looking and seeing no longer have exclusive control over the image, which is instead the pivot of a series of performative, discursive, political, technological, etc. practices. More than objects of representation, images can be described as sensitive and operational interfaces, as environments to be experienced and crossed, as participatory infrastructures. This perspective radically challenges the humanities and its research tools. Peter Burke reminds us, for example, how the inclusion of images in historical research means a shift in focus from collecting "sources" to pursuing the "traces" of the past in the present (Burke 2001). Contemporary conspiracy theories, by their very nature hybrid, syncretic, and mutable, cannot be studied with classical methods alone but must be understood precisely in their fragmentary logic, which feeds on selective appropriations and multi-modal communication. This article has given some examples of how this conspiracy genealogy uses historical sources to generate symbols and images as traces for the present. These are only a few partial examples that do not cover the broad spectrum of contemporary political communication and especially global conspiracy mythopoesis. Although some recurring modes and patterns can be detected, it is not possible to predict the forms and arguments that will be used in the near future to keep conspiracy theories, hate speech, and the destabilization of democracies alive and increasingly powerful. Paraphrasing M. Foucault, it could be said that it is necessary to learn to observe "The Disorder of Discourse", all the more so since the relationship between power and language seems increasingly unbalanced in favor of the latter. But, as we have tried to bring out in these pages, contemporary discourses and narratives are not fixed and given once and for all. On the contrary, these discourses and narratives maintain their function and stable core by constantly changing their outlines. And this is possible precisely because of the reproductive and metamorphic capacity of images (ambiguous, hybrid, and adaptable).

Any genealogy of conspiracy thought is thus a genealogy for future use, constructed and reconstructed in order to survive and maintain an aggregative force; thus, to take an example quoted above, the battle between the people of the New World and the Old English World that interested Franklin can be transformed into the conflict between "the real old Americans" (whites) and the new enemy subjectivities (immigrants), turning the terms and values of an ancient and manifestly misunderstood History upside down. In this sense, we must accept that conspiracy thoughts are not shaken by accusations of irrationality, non-adherence to reality, inconsistency, and internal incoherence. These are all accusations that can be heard in a world that admits logical consistency, factual verification, the scientific method in its different disciplines, and the division of argumentative planes as paradigms. The syncretic and unscrupulous creativity of contemporary conspiracies is not an exception or an anomaly but rather the new rule of media and cultural convergence, the most orthodox interpretation of a world that minces and mixes everything to produce novel and illusions of stable meaning in an increasingly complex and disoriented world.

**Funding:** This research received no external funding.

**Institutional Review Board Statement:** Not applicable.

**Informed Consent Statement:** Not applicable.

**Data Availability Statement:** Data are contained within the article.

**Conflicts of Interest:** The author declares no conflict of interest.

## Notes

1    The full speech is available at this link: https://www.c-span.org/video/?532608-1/president-trump-holds-rally-sioux-center-iowa, accessed on 9 May 2024.

2   For a definition of political conspiracies, see Solinas (2023).

3   HIAS (Hebrew Immigrant Aid Society) is a non-profit organisation dedicated to assisting refugees.

4   https://projects.propublica.org/parler-capitol-videos/ accessed on 9 May 2024.

5   From now on, reference will be made to data from conversations taken from the Gab platform with the year in which they took place.

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
