# Peer review of "Multimodal Genealogy: The Capitol Hill Riot and Conspiracy Iconography"

_genealogy, doi:10.3390/genealogy8020058_

Round 1
Reviewer 1 Report
Comments and Suggestions for Authors
The paper is interesting and presents a valuable contribution into the field of semiosis and conspiracy theory. The paper is well structures, however, a final discussion or a sort of conclusion is missing. I suggest to authors to add an additional section at the end.
Author Response
Thank you for the precious feedback, which I agree with and which helps to improve the article. In order not to exceed the number of words for the article I cut the part dedicated to methodology and research data as well as the concluding analysis. I have tried to briefly integrate these two points by inserting a summary indication on the corpus at the end of paragraph 2.1 and the conclusion notes at the end of the article (in yellow).

Reviewer 2 Report
Comments and Suggestions for Authors
This paper offers a rigorous interrogation of the multimodal forms of communication represented by iconography and discourse surrounding the Capital Hill riot.
It offers important insight into an inflection point in American democracy and political communication.
The paper could more clearly state its unique contribution in the introduction. Plus, its research questions need to come at the end of the theoretical section.
The paper could include a sample size. That is, how many images, symbols and online discussion platforms comprise the corpus?
Also, how was the corpus drawn? What was the selection process? What was the criteria for including and excluding materials for analysis?
Most importantly, the analysis should return to the theoretical-mythological framework outlined at the start. How does this paper’s analysis intersect with methodology and theory? What do we learn from this analysis about the key thinkers identified in the literature review? For instance, what do we learn about theories of representation (Curtis 2010; Mitchell 2013) and discursive dimensions (Foucault 1971)? How does this paper’s analysis push or test theory?
Also, the conclusion requires some reflexivity about this contribution’s limitations. Furthermore, where does future research go after this study?
As well, the conclusion would benefit from some synthesis of how researchers can use this study’s findings as a theoretical-methodological framework.
Thanks for a well written and engaging argument.
Author Response

(The authors gave the same response as above.)

Round 2
Reviewer 2 Report
Comments and Suggestions for Authors
Thanks for addressing my suggestions.